# Integrating human papillomavirus testing as a point-of care service using GeneXpert platforms: Findings and lessons from a Kenyan pilot study (2019–2020)

**Valerian Mwenda** [1]*, **Joan-Paula Bor**[1], **Mary Nyangasi**[1], **James Njeru**[2], **Sharon Olwande**[2], **Patricia Njiri**[2], **Marc Arbyn**[3,4], **Steven Weyers**[4,5], **Philippe Tummers**[4,5,6], **Marleen Temmerman**[5,7]

1 National Cancer Control Program, Ministry of Health, Nairobi, Kenya, 2 Clinton Health Access Initiative, Nairobi, Kenya, 3 Unit of Cancer Epidemiology, Belgian Cancer Centre, Sciensano, Brussels, Belgium, 4 Department of Human Structure and Repair, Gent University Hospital, Ghent, Belgium, 5 Department of Obstetrics and Gynaecology, University Hospital, Gent, Belgium, 6 Cancer Research Institute Ghent (CRIG), Ghent, Belgium, 7 Department of Obstetrics and Gynaecology, Aga Khan University Hospital, Nairobi, Kenya

* valmwenda@gmail.com

**Data Availability Statement:** Data cannot be shared publicly because it is subject to the national public health surveillance data regulations. Data are

## Abstract

### Background

Globally, cervical cancer is a major public health problem, with about 604,000 new cases and over 340,000 deaths in 2020. In Kenya, it is the leading cause of cancer deaths, with over 3,000 women dying in 2020 alone. Both the Kenyan cancer screening guidelines and the World Health Organization's Global Cervical Cancer Elimination Strategy recommend human papillomavirus (HPV) testing as the primary screening test. However, HPV testing is not widely available in the public healthcare system in Kenya. We conducted a pilot study using a point of care (POC) HPV test to inform national roll-out.

### Methods

The pilot was implemented from October 2019 to December 2020, in nine health facilities across six counties. We utilized the GeneXpert platform (Cepheid, Sunnyvale, CA, USA), currently used for TB, Viral load testing and early infant diagnosis for HIV, for HPV screening. Visual inspection with acetic acid (VIA) was used for triage of HPV-positive women, as recommended in national guidelines. Quality assurance (QA) was performed by the National Oncology Reference Laboratory (NORL), using the COBAS 4800 platform (Roche Molecular System, Pleasanton, CF, USA). HPV testing was done using either self or clinician-collected samples. We assessed the following screening performance indicators: screening coverage, screen test positivity, triage compliance, triage positivity and treatment compliance. Test agreement between local GeneXpert and central comparator high-risk HPV (hrHPV) testing for a random set of specimens was calculated as overall concordance and kappa value. We conducted a final evaluation and applied the Nominal Group Technique (NGT) to identify implementation challenges and opportunities.

available from the National Cancer Control Program, Ministry of Health, Committee (contact via headnccp@gmail.com) for researchers who meet the criteria for access to confidential data.

**Funding:** This work was supported by funding from Unitaid through Clinton Health Access Initiative (grant ID UNITCANCER1). The funders had no role in study design, data collection and analysis, decision to publish, or preparation of the manuscript.

**Competing interests:** The authors have declared that no competing interests exist.

## Key findings

The screening coverage of target population was 27.0% (4500/16,666); 52.8% (2376/4500) were between 30–49 years of age. HPV positivity rate was 22.8% (1027/4500). Only 10% (105/1027) of HPV positive cases were triaged with VIA/VILI; 21% (22/105) tested VIA/VILI positive, and 73% (16/22) received treatment (15 received cryotherapy, 1 was referred for biopsy). The median HPV testing turnaround time (TAT) was 24 hours (IQR 2–48 hours). Invalid sample rate was 2.0% (91/4500). Concordance between the Cepheid and COBAS was 86.2% (kappa value = 0.71). Of 1042 healthcare workers, only 5.6% (58/1042) were trained in cervical cancer screening and treatment, and only 69% (40/58) of those trained were stationed at service provision areas. Testing capacity was identifed as the main challenge, while the community strategy was the main opportunity.

## Conclusion

HPV testing can be performed on GeneXpert as a near point of care platform. However, triage compliance and testing TAT were major concerns. We recommend strengthening of the screening-triage-treatment cascade and expansion of testing capacity, before adoption of a GeneXpert-based HPV screening among other near point of care platforms in Kenya.

## Background

In 2020, an estimated 604,000 cases of cervical cancer were diagnosed globally and more than 340,000 women died from the disease [1]. Africa contributes to approximately a quarter of global deaths from cervical cancer [2]. Although cervical cancer screening programs have been successful in reducing the cervical cancer burden in most high-income countries, many countries in sub-Saharan Africa (SSA) have not been able to establish and sustain cervical cancer screening programs at the population level due to financial, logistical and socio-cultural barriers, among other challenges. While 21% of SSA countries have organized cervical cancer screening programs, only 4% have screening at population level and only 2% use human papillomavirus (HPV) testing as a screening approach [3]. Embarrassment and stigma have been identified as key individual barriers to cervical cancer screening uptake in SSA [4–6]. Lack of programmatic cervical cancer screening, either due to competing national priorities or cultural attitudes have also contributed to the low screening coverage rates in SSA [3].

Cervical cancer screening using HPV testing has been shown to protect more effectively against cervical cancer than cytology or VIA [7–9]. As part of the global cervical cancer elimination strategy, the World Health Organization (WHO) advises countries to adopt HPV testing as the high precision screening modality, at age 35 and 45 years [10]. However, the effective adoption of HPV-based cervical cancer screening at the population level in SSA has to contend with several challenges. First, there is the need for adequate funding for health system strengthening, especially laboratory capacity to undertake HPV testing [11]. Second, is to assure the adequate management of at least 90 percent of the screen-positive women within the current health care system [12]. Third, the continued adherence to evidence-based guidelines is paramount to ensure a high-quality screening program [13, 14].

In 2020, Kenya had an estimated 5,236 cases of cervical cancer (11.9% of all cancers among women), and with over 3,000 deaths annually, it is the leading cause of all cancer deaths [15]. Unfortunately, the uptake of screening for cervical cancer is low at an estimated 16% in 2015

[16]. Existing data from the Kenya Health Facility Assessment of 2018 has shown that only a quarter of eligible healthcare facilities currently offer cervical cancer screening and treatment [17]. The National Cancer Screening Guidelines recommend programmatic screening for all women 30–49 years of age, with HPV DNA testing as the preferred screening test [18]. However, HPV tests are not currently widely available in Kenya. Conducting pilot studies enables policy makers to identify challenges, solutions and opportunities for success before launching national programs [19–23]. We conducted a pilot to assess feasibility of a point-of care HPV testing model utilizing GeneXpert platforms which were already available countrywide for TB and HIV testing programs and validated for primary cervical cancer screening [24], to inform planned national roll-out.

## Methods

### Pilot design and population

The HPV pilot was prospectively implemented from October 2019 to December 2020, in nine health facilities across six counties in Kenya (Fig 1). The pilot utilized GeneXpert machines currently available in sub-county and county hospitals for TB, viral load testing and early infant HIV diagnosis. Selection of counties and facilities was based on availability of cervical cancer screening and treatment services using visual inspection with acetic acid (VIA), cryotherapy and loop electro-surgical procedure (LEEP), availability and level of utilization of GeneXpert platforms for TB and HIV testing with areas recording low utilization levels targeted

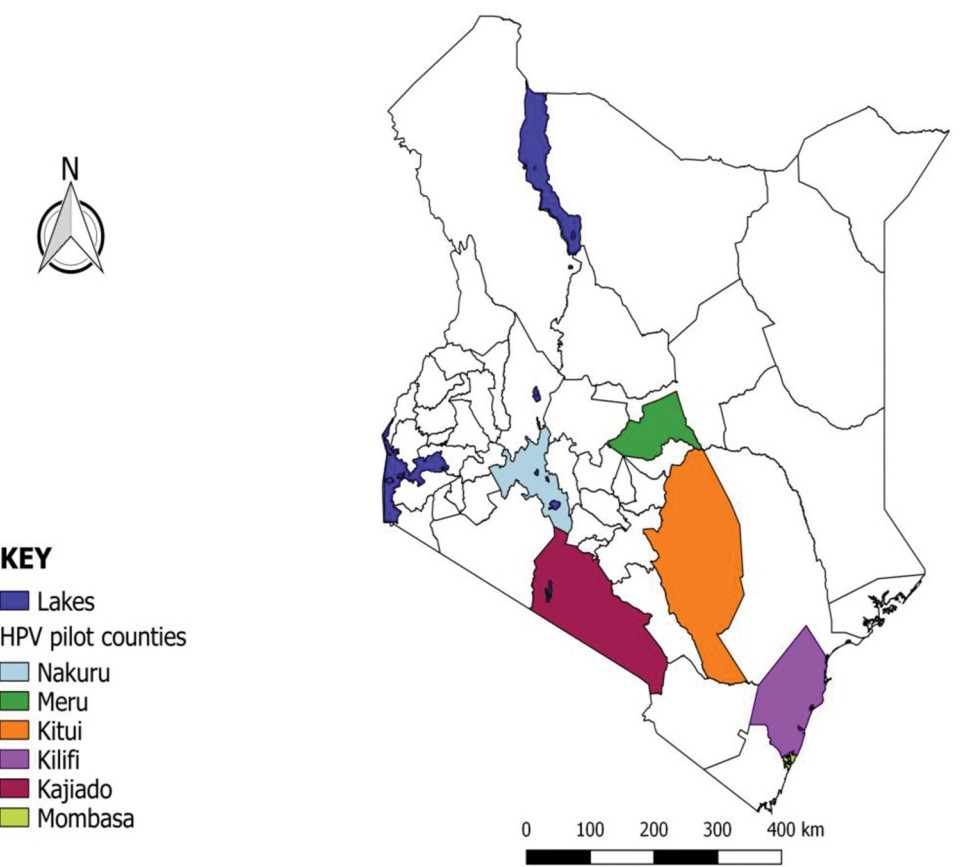

**Fig 1. Map of the selected counties for the HPV testing pilot study in Kenya.**

for the pilot. The target population was women aged 30–49 years. The pilot was implemented through collaboration between the National Cancer Control Program (NCCP), the county departments of health and the Clinton Health Access Initiative (CHAI). The size of the target population was calculated from the 2019 Kenya census, and the figure was 16,666 women.

## Pilot implementation approach

The pilot had three phases: preparation phase (planning, training and procurement of commodities), screening phase and evaluation phase. During the preparation phase, the pilot team mapped the pilot facilities, identified the screening service provision points (outpatient departments, maternal and child health clinics, family planning clinics or comprehensive HIV care clinics) and selected the healthcare workers to be trained. Using pre-developed standardized curricula, we trained both health facility teams (nurses, clinical officers, medical officers, laboratory personnel and data managers) as well as community health workers (CHW). Each training consisted of a two-day didactic session followed by a three-day practicum. The didactic sessions had 12 modules, ranging from physiology and epidemiology of HPV infection to screening, triaging and data management. Training sessions were clustered based on the participating counties. The training content was tailored to the cadre; for instance, CHW were trained using a specially designed cervical cancer screening curriculum for the community level in Kenya, while data managers were only trained on basic concepts of cervical cancer and HPV. HPV sample collection kits, preservative media and GeneXpert HPV cartridges were procured and delivered to the identified facilities and the GeneXpert machines configured to run HPV tests. The second phase involved screening, triage and treatment of eligible women. As per the national cancer screening guidelines, HPV positive women underwent VIA, and those with VIA positive lesions underwent cryotherapy or LEEP at the nine facilities since they had treatment devices. Initially, cervical samples were taken by a health care worker. However, due to the COVID-19 pandemic, a community-based self-sampling approach was introduced to complement specimen collection conducted by the facility health workers. This involved the CHW delivering the testing kits to the women at the community, giving instructions on sample collection, and relaying the sample to the testing facilities. HPV test results were delivered to the women at the health facility by healthcare workers; the CHW only informed the women to visit the health facilities to get the results when ready.

## Quality assurance

Validation of HPV testing on the GeneXpert platform was undertaken by the Kenya Medical Laboratory and Technicians Board (KMLTTB) before the pilot commenced. Throughout the pilot, the National Oncology Reference Laboratory (NORL) conducted quality assurance and assessment covering the nine pilot sites guided by the NORL external quality assurance standard operating procedures (EQA SOP). As part of the quality assurance process, approximately 10% of the samples tested at the pilot sites with known results (HPV16, HPV18, other hrHPV and negative) were sent by courier for retesting on the COBAS Platform at NORL. We tracked the turnaround time (TAT) which is the period between sample collection and relay of results.

## Final pilot evaluation

The final phase of the pilot involved a systematic evaluation, with the following components:

a. Health facility cervical cancer screening human resource mapping: an electronic questionnaire was used to assess the pilot sites on the availability of staff, training capacity, deployment and retention of staff trained for cervical cancer service provision within the facility.

b. Abstraction of cervical cancer screening and treatment data from the Kenya Health Information System augmented by primary cancer screening and laboratory registers.

c. Estimation of the GeneXpert utilization rate: since the platform can run a batch of samples independently on different modules and produce results every two hours, utilization was estimated as the number of tests that were run in an eight-hour working day. For the four module platforms, the maximum utilization was 16 tests while for the 16 module platforms, the number was 64 [25].

d. Nominal group technique (NGT): NGT exercises were conducted with the CHW, staff at the facility screening service points and facility managers. The NGT involved structured brain-storming sessions involving facility managers, cervical cancer screening clinicians, laboratory staff, data managers and CHWs, moderated by the evaluation team. The discussion was centered on the pilot processes, successes, challenges, possible solutions, opportunities and learning points. The process involved ideas generation and recording, discussion around the ideas as per the topic area, ranking of the ideas, review and final identification of key messages on each domain. We conducted at least one NGT exercise for each participating health facility; some had multiple sessions based on the operational level (county or sub-county hospital) and structure of cervical cancer screening service provision.

## Data management

We calculated the following screening indicators: screening coverage, test positivity, proportion of screen-positives triaged, proportion of triage-positive women treated, GeneXpert utilization and invalid sample rates. We also mapped the screening staff attrition at the pilot sites. Test concordance between GeneXpert testing at the pilot sites and COBAS 4800 at NORL were expressed as Cohen's kappa statistic, using the following formula [26]:

$k = (p_o − p_e) / (1 − p_e)$.

Where;

K: Cohen's kappa

$p_o$: Relative observed agreement between the pilot site results and those from NORL.

$p_e$: Hypothetical probability of chance agreement between the two testing sites.

Calculation of the $p_o$ and $p_e$ was based on the dummy table below (Table 1):

Consequently;

$p_o = (a+b)/n$

$p_e = [(n_1/n) * (m_1/n)] + [(n_o/n) * (m_o/n)]$

The interpretation of the kappa statistic was as follows: 0-no agreement; 0.10 to 0.20-slight agreement; 0.21 to 0.40-fair agreement; 041 to 0.60-moderate agreement; 0.61–0.80-substantial agreement; 0.81 to 0.99-near perfect agreement; 1.0-perfect agreement. Statistical analysis was conducted using Stata 17 (StataCorp. College Station, TX). Means were compared using the Student's t-test. A p-value of <0.05 was considered significant for all statistical analyses. Data from the NGT exercises conducted from the nine facilities were combined and top-ranking

**Table 1. Calculation of the kappa statistic.**

| Pilot sites | National oncology reference laboratory | | Total |
| --- | --- | --- | --- |
| | Positive | Negative | |
| Positive | A | b | $m_1$ |
| Negative | C | d | $m_0$ |
| **Total** | $n_1$ | $n_0$ | n |

ideas summarized in tables, based on the various discussion themes, considering both popularity strength and frequency of voting.

## Ethical considerations

These are findings of the final evaluation of a pilot implementation of a public health program. Therefore, data collection was conducted within the national public health surveillance framework. Since this was classified as routine, non-research public health intervention, an institutional Review Board (IRB) process was not necessary. However, both the pilot and the final evaluation were subjected to internal approval processes, by the Ministry of Health and the respective county departments of health.

## Findings

**Screening performance indicators.** Table 2 summarizes the pilot performance statistics. During the 14 month pilot period, 4500 women were screened in the nine facilities, representing 27.0% (4500/16,666) of eligible women in the catchment area of those facilities. Of those screened, 52.8% (2376/4500) were between the ages of 30 and 49 years, 15.6% (700/4500) were below 30 years and 9.7% (438/4500) were 50 years or older; 21.9% (986/4500) had no age information. The overall HPV positivity rate was 22.8% (1027/4500); the positivity had a wide range between the pilot sites, from 7.6% in Port Reitz Sub-county hospital (Mombasa County) to 34.7% at Kilifi County Referral Hospital (Kilifi County). There was no correlation between the testing volumes and invalid sample rate (Pearsons correlation coefficient -0.065, p = 0.87). The median TAT was 24 hours (IQR 2–48 hours).

**HPV testing and linkage to further evaluation and treatment.** Of those screening positive, only 10% (105/1027) had evidence of undergoing triaging using VIA/VILI (Fig 2). For those undergoing VIA/VILI, 21.0% (22/105) tested positive (including 1 case suspicious for cancer). Seventy three percent (16/22) of those positive received either cryotherapy or were successfully referred to the gynecological outpatient clinic (GOPC).

**Cervical cancer screening human resources capacity.** Only 3.5% of all the nurses in the participating facilities were trained in cervical cancer screening and treatment; out of these, 72.4% were offering the service in their facilities at the time of final evaluation of the pilot (Table 3). The numbers were 3.8% and 20.0% for clinical officers, and 15% and 67% for

**Table 2. HPV testing statistics, National HPV testing pilot project, Kenya, 2020.**

| Location | | No. of Gene Xpert modules | Target population coverage | | | Positivity | | Invalid samples | | TAT[a] (hours) |
|---|---|---|---|---|---|---|---|---|---|---|
| County | Testing Site | | Number targeted for screening | Total number screened | % | No. positive | % | No. invalid | % | |
| Nakuru | Molo | 4 | 1,297 | 328 | 25.3 | 80 | 24.4 | 0 | 0.0 | 2 |
| | Naivasha | 4 | 3,337 | 494 | 14.8 | 23 | 4.7 | 2 | 0.4 | 48 |
| Meru | Meru | 16 | 3,704 | 877 | 23.7 | 235 | 26.8 | 3 | 0.3 | 48 |
| Kajiado | Kajiado | 4 | 1,095 | 210 | 19.2 | 23 | 11.0 | 2 | 1.0 | 24 |
| | Loitoktok | 4 | 1,427 | 373 | 26.1 | 87 | 23.3 | 45 | 12.1 | 2 |
| Mombasa | Port Reitz | 4 | 1,355 | 288 | 21.2 | 22 | 7.6 | 0 | 0.0 | 2 |
| | Likoni | 4 | 2,194 | 366 | 16.7 | 46 | 12.6 | 14 | 3.8 | 24 |
| Kilifi | Kilifi | 4 | 1,381 | 1,334 | 96.6 | 463 | 34.7 | 10 | 0.7 | 120 |
| Kitui | Mutomo | 4 | 876 | 230 | 26.3 | 48 | 20.9 | 15 | 6.5 | 96 |
| | **Total** | | **16,666** | **4500** | **27.0** | **1027** | **22.8** | **91** | **2.0** | |

[a]TAT: Turn-around time

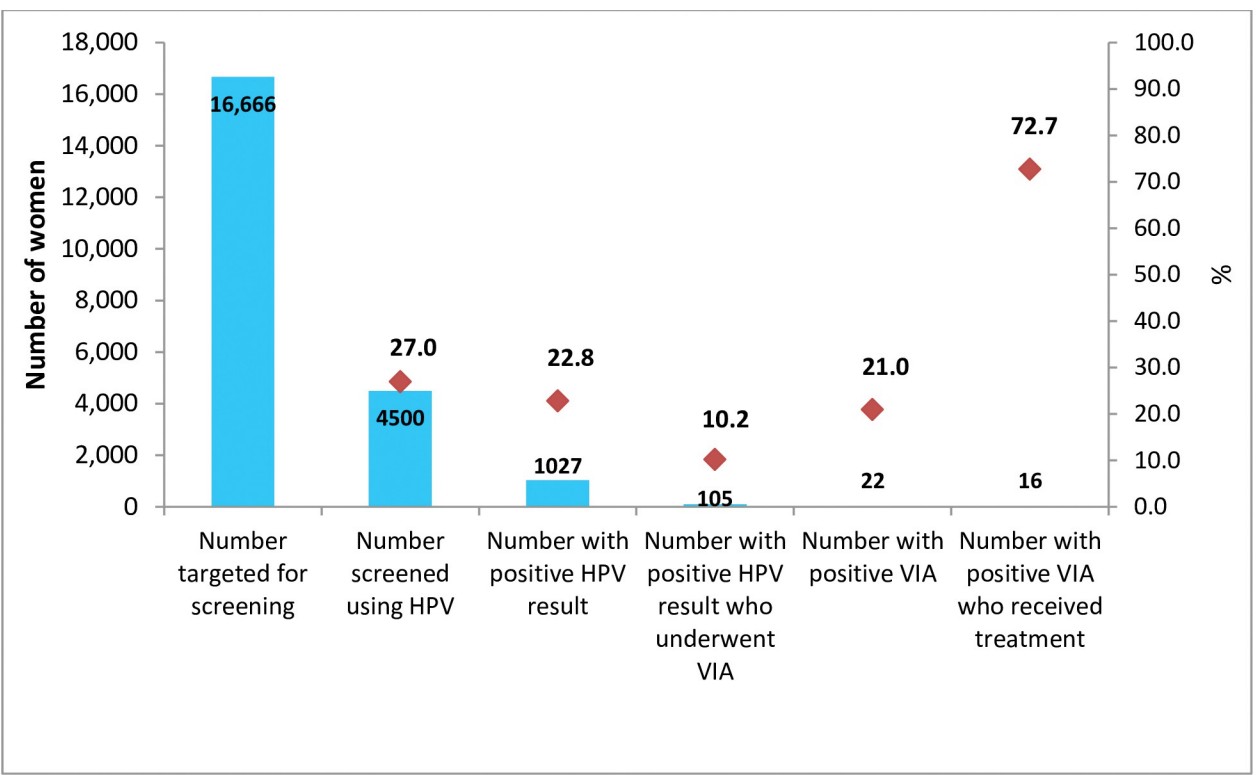

**Fig 2. Care cascade (screening-triaging-treatment) after cervical cancer screening using HPV testing; Kenya HPV screening pilot, 2020.**

medical officers, respectively. Among the 18 gynecologists, 83% were trained and 80% of those trained were offering cervical cancer screening and treatment services.

**Test agreement between Roche COBAS 4800 and Cepheid GeneXpert.** A total of 192 randomly selected samples from the pilot sites were were re-analysed at the NORL, using COBAS 4800 platform as part of the quality assurance process. Out of 129 samples that tested positive at the pilot sites, 26 tested negative at the NORL. All 59 negative samples at pilot sites tested negative at the NORL. Consequently, inter-laboratory agreement between the two pilot sites and the NORL was 86.2% (162/188), while the kappa value was 0.71, translating to substantial level of agreement (Table 4).

**GeneXpert utilization.** Before the pilot, the mean GeneXpert utilization at the pilot facilities was 40.3% (Table 5). During the pilot, the mean utilization was 72.8%, a difference of of

**Table 3. Trained cervical cancer health workforce in the selected counties.**

| Cadre | Total number in the facilities (N) | Number trained in the nine facilities ($n_1$, %)[a] | Number offering cervical cancer screening and treatment ($n_2$,%)[b] |
|---|---|---|---|
| Nurses | 830 | 29 (3.5) | 21 (72.4) |
| Clinical officers | 133 | 5 (3.8) | 1 (20.0) |
| Medical officers | 61 | 9 (14.8) | 6 (66.7) |
| Obstetrics and gynecologists | 18 | 15 (83.3) | 12 (80.0) |

[a]The proportion is calculated as $n_1/N*100$

[b]The proportion is calculated as $n_2/n_1*100$

**Table 4. Test agreement between GeneXpert and COBAS 4800.**

| Pilot sites | National oncology reference laboratory | | |
|---|---|---|---|
| | **Positive** | **Negative** | **Total** |
| Positive | 103 | 26 | 129 |
| Negative | 0 | 59 | 59 |
| Total | 103 | 85 | 188 |

$P_0$ = (103+59)/188 = 0.86
$P_e$ = [(129/188*103/188)] + [(85/188*59/188)] = 0.52
$K$ = (0.86–0.52)/ (1–0.52) = 0.71

32.5% (p = 0.002). Only one facility had a lower utilization rate during the pilot compared with the period before the pilot commenced.

## Successes, challenges and opportunities

During the NGT sessions, self sampling was identifed as a more acceptable collection method; while adopting the community strategy was noted to increase screening uptake, in particular, in situations of health system disruptions mainly resulting from the COVID-19 pandemic (Table 6). The main challenges noted included low awareness at the community, ensuring teamwork between community and facility providers, poor linkage between clinicians and laboratory personnel who run HPV tests (results relay), high loss to follow-up, backlogs, commodity supply chain management and low retention rates of trained human resource personnel at screening sites.

## Discussion

### Summary of findings

We successfully integrated HPV testing into routine GeneXpert at primary care and tertiary facillities in Kenya. Over half of those screened were within the target screening interval, but the overall coverage of the targeted population was low, at 27.0%. Triaging with VIA was low, but linkage to care of VIA-positive women was better at 73.0%. We found long testing TAT and backlogs in most of the testing sites mostly because of competing tests for TB and HIV on machines with testing capacity for only 16 tests in total during the 8-hour shift per day. Substantial agreement between the Cepheid GeneXpert and COBAS 4800 platform was observed. Overall, the pilot increased the utilization of the GeneXpert equipment. On the health system

**Table 5. Change in utilization of GeneXpert equipment during the HPV pilot, Kenya, 2020.**

| County | Pilot facility | GeneXpert utilization (%) | |
|---|---|---|---|
| | | **Before the pilot** | **During the pilot** |
| Nakuru | Molo SCH | 32 | 67 |
| | Naivasha SCH | 60 | 100 |
| Meru | Meru TRH | 58 | 100 |
| Kajiado | Kajiado CRH | 46 | 38 |
| | Loitoktok SCH | 25 | 80 |
| Mombasa | Port Reitz SCH | 28 | 100 |
| | Likoni SCH | 31 | 50 |
| Kilifi | Kilifi CRH | 52 | 80 |
| Kitui | Mutomo SCH | 31 | 40 |
| | **Overall utilization** | **40.3** | **72.8** |

**Table 6. Successes, challenges and opportunities identified during the nominal group discussions, Kenya HPV pilot study, 2020.**

**Successes**

• Self-sample collection approach was more acceptable to the community, especially among communities with strong cultural beliefs and women who did not present in the health facilities.

• By incorporating the community strategy, more women were reached, especially in the context of the COVID-19 pandemic.

• Teamwork was observed between community health volunteers and facility teams (nurses, laboratory officers).

• Screening activity at the HIV comprehensive care centers increased.

• Increased utilization of GeneXpert machine was noted.

**Challenges**

• The screening staff reported difficulties in explaining results when the screened women were found to be HPV positive but VIA/VILI negative.

• Loss to follow-up after a positive HPV result was high: there was no evidence for VIA being done for majority of the women testing HPV positive.

• High turnover of trained health care workers especially among clinical officers: some were either moved to other health facilities or other service provision areas in the same facilities.

• Change resistance was noted in some instances, when new guidelines were introduced (HPV-testing as opposed to VIA as primary screening method).

• Testing capacity and infrastructure: Lack of an efficient laboratory information management system (LIMS) and limited GeneXpert testing capacity, since most were 4-module platforms.

• The high workload of TB and HIV tests on same platforms made it difficult to prioritize HPV testing especially when the platformshad low testing capacity

• Weak clinical- lab interface: suboptimal linkages between clinicians and lab personnel affected sample collection,testing and relay of results.

• Challenges with stock-out of commodities: mostly related to COVID-19 prioritization with resultant disruption of supply chain for HPV commodities

• COVID-19 pandemic caused interruption of cancer screening services at all the facilities

**Opportunities/areas for improvement**

• Increase health education in the community, including targeted outreaches and building capacity of community health workers to deliver targeted health messages on cervical cancer screening using HPV

• Explore and strengthen self collection of HPV samples in the community to complement Health worker assisted sample collection.

• Strengthen the follow-up and care-linkage for HPV positive clients.

• Conduct more training and mentorship programs on screening and treatment of pre-cancer lesions.

• Establish a sustainable framework for referring samples from the commuity to the testing laboratories, for self-collected samples.

• Continous mentorship to strengthen the Lab -clinical interface and optimize work flow in the labs.

• Placement of GeneXpert equipment with capacity to run more tests at a time such as the 16 module platforms

• Utilize roll-out of universal health coverage to support the cervical cancer screening processes.

• Adopt digital health solutions such as use of short messaging services and phone calls to relay results, track patients for follow-up and linkages to care

• Improve both pre-test and post-test counselling as part of the screening process.

• Consider establishing support groups for those with positive HPV results, to improve peer health information exchange and increase follow-up rates.

enablers, inadequate and inefficient deployment and retention of trained healthcare workers and availability of treatment devices was a major limiting factor, while the community strategy was a key opportunity.

## Screening coverage

The pilot was able to reach approximately 27.0% of the targeted number of women in the six counties. This is below the targeted minimum coverage of 70%. One reason may be suboptimal

mobilization and invitation for screening. Although the community strategy was involved in creating awareness and supporting self-sample collection, other approaches such as use of mobile platforms to reach the targeted population and create awareness were not deployed and/or sustained during the pilot. A study in Kenya found that >70% of women reported ownership and daily use of a mobile phone, and 60% indicated that they would be comfortable receiving results via text messaging [27]. Another explanation could be logistical challenges, that caused periodic stock-outs of HPV testing commodities in some centers during the pilot. The COVID-19 pandemic also reduced the overall utilization of health services in Kenya, including cervical cancer screening [28]. In comparision, a recent global analysis of cervical cancer screening coverage estimates global coverage at 32% in the previous five years and 15% in 2019; for SSA, the figures are 15% and 3% respectively [29]. Slightly over half of the women screened fell within the programmatic target HPV screening age of 30–49 years. For successful population level screening programs, the WHO recommends that at least 70% of those screened should be within the target screening age bracket [30].

## HPV positivity and linkage to care

The overall high risk HPV positivity rate was 22.8%. This is within the target range set by the WHO for monitoring HPV-based cervical cancer screening programs (5–25%) [30]. This also compares with HPV positivity rates in Sub-Saharan Africa, of 25.6% (Nigeria), 32.3% (Ghana), 28.5% (South Africa) and 28.2% (Democratic Republic of Congo) [31–34]. Of those who tested positive for high risk HPV subtypes, only 10.0% had evidence of VIA being done. This is much lower than the desired 90% linkage to further evaluation and/or treatment. Our findings in this pilot contrast with that from Rwanda and Cameroon, where 89% and 99% of HPV positive women successfully underwent VIA, respectively [35, 36]. Several reaons may explain this finding. First, it's possible that more women underwent VIA, but documentation in the screening registers was incomplete, especially due to the fact that VIA was performed on a different visit from the initial screening visit. Second, women may have faced economic challenges in seeking care if required to visit a health facility (if screened at the community), return (if screened at the health facility) or referred to a different health facility for triaging. Third, cultural barriers may have played a role in form of the women being denied persmission by their partners to seek further evaluation after detection of a 'sexually transmitted virus.' Lastly, the long testing TAT may also have reduced the likelihood of a successful linkage to triaging and treatment. The VIA positivity for those triaged was 21.0%, which is comparable to studies in Rwanda and Cameroon [20, 36], but higher than findings in Malawi [12]. Of those testing positive on VIA, 73% had evidence of receiving treatment by cryotherapy or being referred to the gynecologic clinic for further management.

## Turn-around time

A wide range from about 2 hours to several days was observed. Samples from the community had longer TAT than facility-collected samples. The main contributing factor to long TAT was prioritization of the GeneXpert platforms for TB and HIV viral load testing and early infant diagnosis as well as low testing capacity of most of the GeneXpert machines with only one site having a 16-module platform. TAT is important in cervical cancer screening because it impacts linkage to care and loss to follow-up. The study from Malawi had a lower TAT of around two hours for sample delivery and a few minutes for sample analysis; however, this study differed from our pilot in that it was a single hospital and surrounding health centers in a hub and spoke model [12].

### Gene Xpert utilization and test agreement with COBAS

The utilization level of the GeneXpert equipment significantly increased during the HPV pilot. While this was a positive finding, we also found that HPV samples were relegated to the background anytime the laboratory had to run TB or HIV samples, which resulted in long TAT in many instances. Most of the pilot facilities had four-module GeneXpert machines which means only about 16 tests could be done on a typical day. While other studies in SSA have also utilized GeneXpert equipment for HPV testing for cervical cancer screening programs [12, 36], our study was unique in that it sought to assess the feasibility of integrating HPV testing into GeneXpert platforms already in use for TB and HIV diagnosis, but with suboptimal utilization. This approach would harness existing capacity within the healthcare system and make implementation more feasible. We found substantial agreement between GeneXpert results and those from COBAS 4800 at the NORL, with a kappa value of 0.71. In comparison, studies by Cuschieri et al and Akbari et al showed a kappa value of 0.91 and 0.93, respectively [37, 38]. The difference could have been caused by the limited number of samples submitted for retesting a the NORL; although the initial plan was to re-test at least 10% of both negative and positive samples, only 4% of the total samples analyzed at the pilot sites was sent for quality assurance, largely due to logisitical difficulties.

### Strengths and limitations

This pilot was implemented within the context of the routine operation of the healthcare system, to mirror what an actual full-scale roll-out would look like. Consequently, the findings are critical in informing planning for wide-scale adoption of GeneXperts and/or any other multiplex platforms that may come to the market in future for cervical cancer screening, through integration with other programs. The main limitation was the logistical challenges, that affected both the implementation of the screening cascade as well the quality assurance processes. Therefore some critical insights from the implementation could have been missed due to incomplete application of some processes.

## Conclusion and recommendations

The community strategy can be an effective resource for community education and supporting self-sample collection at the community level, with modest investment and support. Routinely incorporating self-sample collection can increase screening uptake though it is prone to errors therefore requires a proper quality assurance system to be in place. Communication of results and linkage to care and follow-up requires use of innovative m-health startegies as well as a properly structured health information system, to ensure an effective fail-safe mechanism. For the HPV test to be used as a point-of-care test for cervical cancer screening, high-volume sites should be considered for upgrade to higher throughput GeneXpert machines or placement of any new and innovative multiplex platforms that may come into the market in future. A better coordination between the community, sample referral mechanism and the testing laboratories can reduce the TAT. Integrating HPV sample referral into the existing TB and HIV networks may help in improving efficiencies but could require more investment in areas with suboptimal sample referral networks. A laboratory information management system (LIMS) can enhance processing and transmission of test results, reduce transcription errors and enhance quality improvement.

## Acknowledgments

We wish to acknowledge the various County Departments of Health and health facility teams who were instrumental in the implementation and evaluation of the HPV screening pilot. We also acknowledge the NORL, for their role in the QA processes for the pilot.

## Author Contributions

**Conceptualization:** Valerian Mwenda, Joan-Paula Bor, Mary Nyangasi, Sharon Olwande, Patricia Njiri.

**Data curation:** Valerian Mwenda, James Njeru.

**Formal analysis:** Valerian Mwenda.

**Funding acquisition:** Mary Nyangasi, Patricia Njiri.

**Investigation:** Joan-Paula Bor, Sharon Olwande, Patricia Njiri.

**Methodology:** Valerian Mwenda, Sharon Olwande, Steven Weyers, Marleen Temmerman.

**Project administration:** Joan-Paula Bor, Mary Nyangasi, Sharon Olwande, Patricia Njiri.

**Resources:** Patricia Njiri.

**Supervision:** Joan-Paula Bor, Mary Nyangasi, Sharon Olwande, Patricia Njiri.

**Validation:** James Njeru, Marc Arbyn, Steven Weyers, Philippe Tummers, Marleen Temmerman.

**Writing – original draft:** Valerian Mwenda.

**Writing – review & editing:** Valerian Mwenda, Joan-Paula Bor, Mary Nyangasi, James Njeru, Sharon Olwande, Patricia Njiri, Marc Arbyn, Steven Weyers, Philippe Tummers, Marleen Temmerman.

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
