## [Decision Letter · Decision Letter 0]

23 Feb 2023

PONE-D-22-26910Integrating human papilloma virus testing as a point-of care service using GeneXpert platforms: Findings and lessons from a Kenyan pilot study (2019-2020)PLOS ONE

Dear Dr. Mwenda,

Thank you for submitting your manuscript to PLOS ONE. After careful consideration, we feel that it has merit but does not fully meet PLOS ONE’s publication criteria as it currently stands. Therefore, we invite you to submit a revised version of the manuscript that addresses the points raised during the review process.

We look forward to receiving your revised manuscript.

Kind regards,

Armando Baena-Zapata, Ph.D.

Academic Editor

PLOS ONE

2.We note that you have indicated that data from this study are available upon request. PLOS only allows data to be available upon request if there are legal or ethical restrictions on sharing data publicly. For more information on unacceptable data access restrictions, please see http://journals.plos.org/plosone/s/data-availability#loc-unacceptable-data-access-restrictions.

Reviewers' comments:

Reviewer's Responses to Questions

**Comments to the Author**

1. Is the manuscript technically sound, and do the data support the conclusions?

Reviewer #1: Yes

Reviewer #2: Yes

2. Has the statistical analysis been performed appropriately and rigorously? 

Reviewer #1: Yes

Reviewer #2: Yes

3. Have the authors made all data underlying the findings in their manuscript fully available?

Reviewer #1: Yes

Reviewer #2: Yes

4. Is the manuscript presented in an intelligible fashion and written in standard English?

Reviewer #1: Yes

Reviewer #2: Yes

5. Review Comments to the Author

Reviewer #1: Title: Integrating human papilloma virus testing as a point-of care service using GeneXpert platforms: Findings and lessons from a Kenyan pilot study (2019-2020)

The aim of this pilot study was to assess feasibility of a point-of care HPV testing model utilizing GeneXpert platforms which were already available countrywide for TB and HIV testing programs and validated for primary cervical cancer screening to inform planned national roll-out.

This study addresses an important topic and highlights an important issue regarding access to screening, diagnosis, and treatment for cervical cancer prevention in low resources settings.

Minor comments

Methods

In line 143, authors mentioned that population was calculated from the 2019 Kenya census. I suggest that the authors may include total number of women in this section.

Pilot implementation approach: Authors mentioned that the pilot had three phases. I think the authors should include some further detail regarding these phases. This information could improve the methods section. For example, it could be useful a deeper description about training (e.g., which content was included in training sessions? How many training sessions were carried out?); community-based self-sampling approach (Who offered the Self-collection? Where they offered it? Did they use any communication material to explain how to do self-collection?); Delivery of results (Who explain HPV-results and the following steps? The personnel in charge of delivery of HPV- results had received training about it?

Regarding evaluation phase more details regarding NGT exercises are also needed (for example framework, methodology, etc.). In Analysis section they could include details regarding qualitative analysis of data from the NGT exercises.

Results

Screening and follow up indicators: It could be useful, if it is possible, that authors report screening and follow up percentages among women who perform clinician collected and self-collected samples. Adherence to follow up could be lower among women with self-collected samples than among women who attend to health centers to be screened.

Discussion

Authors pointed out that “only 10.0% had evidence of VIA being done”. They mentioned that a possible explanation may be that more women underwent VIA, but documentation in the screening registers was incomplete, especially due to the fact that VIA was performed on a different visit from the initial screening visit”. Maybe, if the VIA was performed on a different visit from the initial screening, it could be possible that other institutional, subjective or economic barriers could be explain the low adherence rate. Evidence regarding follow up adherence showed that women in low-middle resource setting face a lot of institutional, cultural or economic barriers that reduce adherence to follow up. I suggest that authors could discussed some of these barriers or explain better why the considerer that it is due of lack of registration.

In the manuscript it is not clear how the health providers delivered results. In fact, one of challenges reported was that the screening staff had difficulties in explaining results. Evidence have shown that HPV positivity can have an important negative impact on the psychosocial health of tested women. In addition, for many women, information provided by health providers about HPV is often confusing; women have difficulties in understanding HPV-test results and steps to follow. Both HPV’s psychosocial impact of HPV-testing, and lack of information regarding the follow-up process might not only diminish women's quality of life but also reduce their retention to follow up. Other important point to explain the low follow up rate could be the wide range of TAT.

I suggest that authors may consider these points in Discussion section.

In line 376, remove dot: “placement of any new and innovative multiplex platforms that may come into the market in future.. A better”

Reviewer #2: Integrating human papilloma virus testing as a point-of care service using GeneXpert platforms: 2 Findings and lessons from a Kenyan pilot study (2019-2020)

Some observations:

Abstract

Key Findings:

line 61: is 52.8% instead 53.8% please review

Data management

Page 15: lines 201 and 202: table is not referenced in text.

Findings

Screening performance indicators Page 16:

Line 231: Table 1: HPV testing statistics, National HPV testing pilot project, Kenya, 2020: is not referenced in text.

Gene Xpert utilization

Page 19: Table 4: Change in utilization of gene Xpert equipment during the HPV pilot, Kenya, 2020: is not referenced in text.

Page 19: Successes, challenges and opportunities

Table 5: Successes, challenges and opportunities identified during the nominal group discussions, Kenya HPV pilot study, 2020: is not referenced in text.

Line 277: is pandemic? Or panemic?

Discussion

Page 23, line 364 is implementation

Conclusion and recommendations

Page 24 line 376: …. After future only a dot.

6. PLOS authors have the option to publish the peer review history of their article (what does this mean?). If published, this will include your full peer review and any attached files.

Reviewer #1: **Yes: **Melisa Paolino

Reviewer #2: No

---

## [Author Response · Author response to Decision Letter 0]

24 Mar 2023

Our detailed response to reviewers has been uploaded as a file.

---

## [Editor Report · Decision Letter 1]

11 May 2023

Integrating human papilloma virus testing as a point-of care service using GeneXpert platforms: Findings and lessons from a Kenyan pilot study (2019-2020)

PONE-D-22-26910R1

Dear Dr. Mwenda,

We’re pleased to inform you that your manuscript has been judged scientifically suitable for publication and will be formally accepted for publication once it meets all outstanding technical requirements.

Kind regards,

Armando Baena, Ph.D.

Academic Editor

PLOS ONE

---

## [Editor Report · Acceptance letter]

18 May 2023

PONE-D-22-26910R1 

Integrating human papillomavirus testing as a point-of care service using GeneXpert platforms: Findings and lessons from a Kenyan pilot study (2019-2020) 

Dear Dr. Mwenda:

I'm pleased to inform you that your manuscript has been deemed suitable for publication in PLOS ONE. Congratulations! Your manuscript is now with our production department. 

Kind regards, 

on behalf of

Dr. Armando Baena-Zapata 

Academic Editor

PLOS ONE